

# Forecasting tropical cyclone tracks in the Northwest Pacific based on a deep-learning model

Liang Wang[1,2], Bingcheng Wan[3], Shaohui Zhou[3], Haofei Sun[1,2], Zhiqiu Gao[1,3*]

[1] State Key Laboratory of Atmospheric Boundary Layer Physics and Atmospheric Chemistry, Institute of Atmospheric Physics, Chinese Academy of Sciences, Beijing, 100029, China

[2] University of Chinese Academy of Sciences, Beijing, 100049, China

[3] School of Atmospheric Physics, Nanjing University of Information Science and Technology, Nanjing, 210044, China

*Correspondence to*: Dr. Zhiqiu Gao(zgao@mail.iap.ac.cn)

**Abstract.** Tropical cyclones (TCs) are one of the most severe meteorological disasters, making rapid and accurate track forecasts crucial for disaster prevention and mitigation. Because TC tracks are affected by various factors (the steering flow, thermal structure of the underlying surface, and atmospheric circulation), their trajectories present highly complex nonlinear behavior. Deep learning has many advantages in simulating nonlinear systems. In this paper, we explore the movement of TCs in the Northwest Pacific from 1979 to 2021 based on deep-learning technology, divided into training (1979–2014), validation (2015–2018), and test sets (2019–2021), and create 6–72 h TC track forecasts. Only historical trajectory data are used as input for evaluating the forecasts of the three recurrent neural networks utilized: recurrent neural network (RNN), long short-term memory (LSTM), and gated recurrent unit (GRU) models. The GRU approach performed best; to further improve forecast accuracy, a model combining GRU and a convolutional neural network (CNN) called GRU_CNN is proposed to capture the characteristics varying with time. By adding reanalysis data of the steering flow, sea-surface temperatures, and geopotential height around the cyclone, we can extract sufficient information on the historical trajectory features and three-dimensional spatial features. The results show that GRU_CNN outperforms other deep-learning models without CNN layers. Furthermore, by analyzing three additional environmental factors through control experiments, it can be concluded that the historical steering flow of TCs plays a key role, especially for short-term predictions within 24 h, while sea-surface temperatures and geopotential height can gradually improve the 24–72-h forecast.



The average distance errors at 6 h and 12 h are 17.22 km and 43.90 km, respectively. Compared with
the forecast results of the Central Meteorological Observatory, the model proposed herein is suitable
for short-term forecasting of TC tracks.
**1 Introduction**
The Northwest Pacific is the most active basin for tropical cyclones (TCs) in the world, generating over
one-third of the total number of TCs (Gray, 1968). China, located on the western side of the Pacific
Ocean with a coastline longer than 18,000 km, is one of the countries most severely influenced by TCs.
These storm systems are accompanied by strong winds, heavy precipitation, and storm surges, resulting
in severe disasters that affect human lives and economic growth (Goldenberg et al., 2001). Studies have
shown that global warming will progressively intensify TCs over time (Emanuel, 2017; Schulthess et
al., 2019). Since disasters caused by TCs are unavoidable and potentially destructive, accurately
predicting the movement of TCs can provide sufficient preparation time for people in affected areas to
implement disaster mitigation strategies.
Given the uncertainty of TC movements, the complexity and nonlinearity inherent in the
atmospheric system, and the scarcity of ocean-based observational data, accurately predicting the
center positions and intensities of TCs is a challenge. Currently, forecasting methods for TCs are
mainly divided into two categories, with the primary method being numerical weather prediction
(NWP). NWP calculates the approximate solution of partial differential equations involving
atmospheric state variables when the initial conditions and boundary conditions of the atmosphere are
known. In this way, some elements, such as the tracks and intensities of the TCs, can be solved
iteratively; GRAPES-TYM (CMA), GFS (NCEP), and IFS (ECWMF) are the main NWP models.
Although these model forecasts can provide accurate results, there are limitations in methods requiring
numerous calculations, accurate mathematical descriptions of physical atmospheric mechanisms, and
precise initial conditions. At the same time, ensemble forecast methods (GRAPES-GEFS,
ECMWF-EPS, NCEP-GEFS) have been used to reduce the influence of various uncertainties on the
numerical prediction results (Goerss, 2000). The other forecasting method is a statistical model, which
generally utilizes multiple regression. The statistical model is mainly based on the relationship between
the movement of the TC and its specific historical characteristics, but it usually does not consider any



physical processes. The National Hurricane Center has successively adopted statistical models such as
NHC64 (taking observational data and historical 12h movements as factors), NHC67 (increasing
factors based on NHC64), CLIPER (climate persistence factors) (Neumann and Hope, 1972), and
NHC72 (a combination of NHC67 and CLIPER). Most traditional TC statistical models adopt a linear
regression model, and it is difficult for this approach to address the nonlinear problems in TC track
forecasting (Roy and Kovordányi, 2012). At the same time, manual feature selection is unable to
produce accurate predictions.

Deep learning is an emerging application of supercomputing that is continuously being developed;

many researchers have tried to adopt this technology to forecast weather and meteorological elements,
including visibility (Ortega et al., 2022), wind speeds (Liu et al., 2018), radar echoes (Klein et al.,
2015), and precipitation nowcasting (Shi et al., 2015). Deep learning is a statistical model that solves
nonlinear and complex relationships from historical sample data based on neural network algorithms.
The weight factor between network nodes is automatically adjusted through repeated training; thus,
neural network algorithms have the advantages of strong adaptability and fault tolerance. TCs have
complex dynamic mechanisms and are easily affected by many factors, including environmental
steering flow, Beta effects, underlying surface conditions, the asymmetric structure of the inner core,
and mesoscale circulations (Chan and Kepert, 2010). Artificial neural networks (ANNs) have been
applied to predict TC tracks due to their strong learning ability and advantages in simulating nonlinear
systems. Until the 2010s, ANN and back propagation (BP) networks were the mainstream neural
network methods for forecasting TC tracks (Ali et al., 2007; Li-Min et al., 2009; Wang et al., 2011).
Since the mid-2010s, more new methods have been introduced into TC prediction due to the
development of deep-learning technology. Recurrent neural networks (RNNs) are suitable for TC track
forecasting owing to their ability to handle time series data of arbitrary lengths. Moradi Kordmahalleh
et al. (2015)applied a sparse RNN to Atlantic hurricane trajectory prediction using the dynamic time
warping (DTW) method to measure the hurricane most similar to the target hurricane for training. Gao
et al. (2018) used long short-term memory (LSTM) to predict typhoon tracks in the Northwest Pacific
Ocean; the ratio of the cyclone training set and test set was set at 8:2, and the 24-h prediction error
could reach 105 km. Alemany et al. (2018) proposed an RNN based on a grid system to predict
hurricanes in the Atlantic, potentially improving the 6-h prediction accuracy with a root mean square





error (RMSE) of 0.11 for the test set. Kim et al. (2018) performed a TC identification task based on
ConvLSTM to train WRF-simulated data, and the results are significantly better than those of a
convolutional neural network (CNN). These CNNs have attracted attention given their suitability for
processing 2D image data; they maintain spatial correlations by implementing convolution layers and
then pooling layers for feature extraction. Giffard-Roisin et al. (2020) combined historical trajectory
data with wind field reanalysis data as input to a CNN and predicted Atlantic hurricane tracks since
1979, with an average error of 32.9 km for 6-h predictions.

Making full use of different types of data is essential for deep learning. TC-related data are mainly

divided into the following three categories: observational trajectory data, remote sensing data, and
meteorological reanalysis field data. A multi-modal approach enables more accurate predictions than
an approach using a single data source does. Zhang et al. (2018) developed a matrix neural network
(MNN) model that preserves the spatial information of the TC tracks, and it has demonstrated the
ability to provide more accurate results compared with other models (GRU, LSTM, MLP, and RNN).
Ruttgers et al. (2019) built generation adversarial networks (GANs) adding satellite images to predict
the coordinates of the typhoon center and generate cloud maps of future typhoons. Liu et al. (2022)
proposed a new deep learning–based model, DBFNet, to effectively fuse the inherent features of
cyclones and reanalyze 2D pressure field data. Previous studies have shown that deep-learning models
that incorporate multiple data types can improve the track forecast of TCs to a certain extent. Still, most
of them have neglected to describe and analyze the meteorological factors that affect the movement of
TCs, ignoring valuable features.

Therefore, in this paper, we propose a new method for TC track prediction based on a

combination of CNN and GRU models that incorporate data regarding the trajectory, steering airflow,
sea-surface temperatures, and geopotential height as input features, aiming to improve the accuracy of
TC track forecasts by leveraging big data. The main contents of this paper are as follows: Section 2
introduces the necessary data and methodology principles. Section 3 describes the experimental design
and the framework of the fusion model (GRU_CNN) proposed in this paper. Section 4 presents the
experimental results and comparative analysis, and Section 5 provides a summary and discussion of
shortcomings and directions for future work.



## 2 Data and methods

### 2.1 Data

The data used in this paper are trajectory data and reanalysis environmental data. The TC track data come from the International Best Track Archive for Climate Stewardship (IBTrACS), which encompasses all TCs globally. For each TC, the latitude, longitude, central pressure, maximum wind speed, direction, moving speed, and other data are recorded at 3-h intervals. The IBTrACS dataset contains data from different basins where cyclones show different characteristics; thus, this paper only selects TCs that occur in the Northwest Pacific Ocean. To better mine the hidden information, the 2D cyclone track data were chosen according to the method proposed by Li-Min et al. (2009), and 19 movement characteristics were obtained, including the past 24-h longitude, latitude, central atmospheric pressure, maximum wind speed, meridional moving speed, zonal moving speed, moving direction/speed, the difference between those values and those at the current time, and the angle, zonal distance, and meridional distance formed between the data over the past 24 h and in the present moment. In addition, the Coriolis parameter corresponding to the latitude of the past 24 h influences the geostrophic deflection force on the TCs.

Both observational and theoretical studies have shown that TC movement is closely related to large-scale airflow fields (Holland, 1983), and TC movement is mainly affected by the steering flow (Brand et al., 1981; Chan, 1984). Because they are influenced by the earth's rotation, TCs will be biased to the northwest (Kitade, 1981). Interactions among weather systems, the subtropical anticyclone, Westerlies, and the Tibetan High will also affect the movement of cyclones (George and Gray, 1976; Chan et al., 1980).

The geopotential heights of 300 hpa, 500 hpa, and 700 hpa are selected as the locations for the high, middle, and low-level circulation data, respectively. In addition, the underlying surface conditions must be considered, and, in the case of a weak guidance environment, TCs tend to move toward warmer sea-surface temperatures (Sun et al., 2017; Katsube and Inatsu, 2016). Meteorological environmental data are obtained by downloading high-resolution ERA5 reanalysis data from the European Centre for Medium-Range Forecasting (ECMWF). Holland (1984) noted that the deep mean



circulation from 850 hpa to 300 hpa can better represent the direction of a TC. Therefore, the
environmental data for the preceding 24 h were extracted as follows:
(1) The $u$- and $v$-component data of the wind field on the four isobaric surfaces (300 hpa, 500 hpa, 700
hpa, 850 hpa): A 10° radius is extended from the center of the TC. Since the resolution of the selected
reanalysis data is 1° × 1°, a 21 × 21 grid can be formed.
(2) The sea-surface temperature (SST): A 10° radius is again extended from the TC center to form a 21
× 21 grid.
(3) The geopotential heights of 300 hpa, 500 hpa, and 700 hpa: A grid is extended +35° to the north,
−10° to the south, −40° to the west, and +40° to the east from the center of the TC, forming a 46 × 81
grid.
Because the actual weather circulation is very complex and includes information about the TC itself,
the surrounding airflow, and the interaction between the two, it is necessary to separate the cyclone
vortex from the surrounding airflow to obtain the steering flow. The most commonly used method
(Lownam, 2001; Galarneau and Davis, 2013) corrects the vorticity and divergence by solving the
change in the velocity stream and potential functions, respectively, and then calculates the modified
velocity field. The modified flow field can be interpreted as having a non-rotating wind and
non-diverging wind. There must be potential velocity in the irrotational motion and a stream function
in the non-divergent motion. The relationship between them can be expressed as follows:
$$\nabla^2 \psi = \zeta \tag{1}$$

$$v_\psi = \hat{k} \times \nabla \psi \tag{2}$$

Where $\psi$ is the stream function without divergence, $\zeta$ is the relative vorticity, and $v_\psi$ is the
non-divergent wind (rotating wind). To define the rotating wind, the vorticity outside the vortex radius
is set to zero, and $\psi = 0$ is specified on the horizontal boundary. The iterative relaxation method is used
to solve the stream function of Eq. (1) at all layers and then to calculate $v_\psi$ using Eq. (2). In the case of
divergence, Eqs. (1) and (2) are replaced by:
$$\nabla^2 \chi = \delta \tag{3}$$

$$v_\chi = \nabla \chi \tag{4}$$

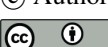



Where $\chi$ is the potential velocity, $\delta$ is the divergence, and $v_\chi$ is the non-vorticity wind. The divergence
outside the vortex radius is set to zero and the potential function $\chi = 0$ on the boundary of the region.
The velocity potential can be solved in the same manner to calculate $v_\chi$. The ambient wind field with
the vortex removed can be obtained by subtracting the rotating wind and divergent wind from the
original wind field, V:
$$v_{env}(x, y, p) = V(x, y, p) - v_\chi(x, y, p) - v_\psi(x, y, p) \qquad (5)$$

**2.2 Methods**
**2.2.1 Random forest**
Selecting features based on importance is a fundamental step in machine learning that most efficiently
directs variables for machine learning models (Díaz-Uriarte and Alvarez De Andrés, 2006; Genuer et
al., 2010). Random forest is a supervised learning method that selects the best feature combination and
reduces the input feature dimension. The random forest contains N decision trees, and N is generally
set to 100. Since bootstrapping (random sampling with replacement) is used to generate the random
decision tree, all samples are not in the generation process of a tree, and the unused samples are called
"out-of-bag" (OOB) samples. Through OOB samples, the accuracy of this tree can be evaluated, and it
is usually calculated via the following four steps:
(1)   Select the OOB data and calculate the OOB data error:
$$\text{error} = \frac{1}{n} \sum_{i=1}^{n} (f_i - y_i)^2 \qquad (6)$$

$f_i$: predicted value, $y_i$: actual value, n: number of samples (1/3 of the total).
(2)   Randomly add noise interference to the features of all samples of the OOB data and calculate the
OOB data error ($error^m$) again.
$$\text{error}^m = \frac{1}{n} \sum_{i=1}^{n} (f_i^m - y_i)^2 \qquad (7)$$

$f_m$: Predicted value after adding noise to feature m.
(3)   Calculate the importance $I$ of all features.




$$I = \frac{1}{N}\sum_{j=1}^{N}(\text{error}_{j}^{m} - \text{error}_{j}) \qquad (8)$$

N: the number of decision trees, $error_j$: OOB data error, $error_j^m$: OOB data error after adding noise.
(4)    According to the order of feature importance, delete the features with less importance in turn,
repeat the above three steps, calculate the OOB data error, and select the combination with the lowest
error. The OOB score represents the score of the model performance:

$$R = 1 - \frac{\sum_{i=1}^{n}(f_i - y_i)^2}{\sum_{i=1}^{n}(y_i - \hat{y})^2} \qquad (9)$$


$$OOB_{\text{score}} = \frac{1}{N}\sum_{j=1}^{N}R_j \qquad (10)$$

Before model training, it is necessary to determine whether the 19 trajectory features all have an impact
on the prediction results. Figure 1(b) shows the 19 features' order of importance calculated using the
random forest. For forecasting the difference in longitude and latitude within the following 72 hours,
characteristics like the historic longitude or the angle formed by the historical moment and the current
moment are significant. The decision about whether to exclude some less important features, however,
requires further consideration. Based on Eq. (9) and Eq. (10), the OOB scores under different input
feature dimensions are computed, with variables input in the order of importance, as shown in Fig. 1(c).
In the case in which the first 11 features are sorted by importance, the OOB score is the highest, and
the features added later will no longer affect the result; in other words, the best combination is that of
the first 11 features.



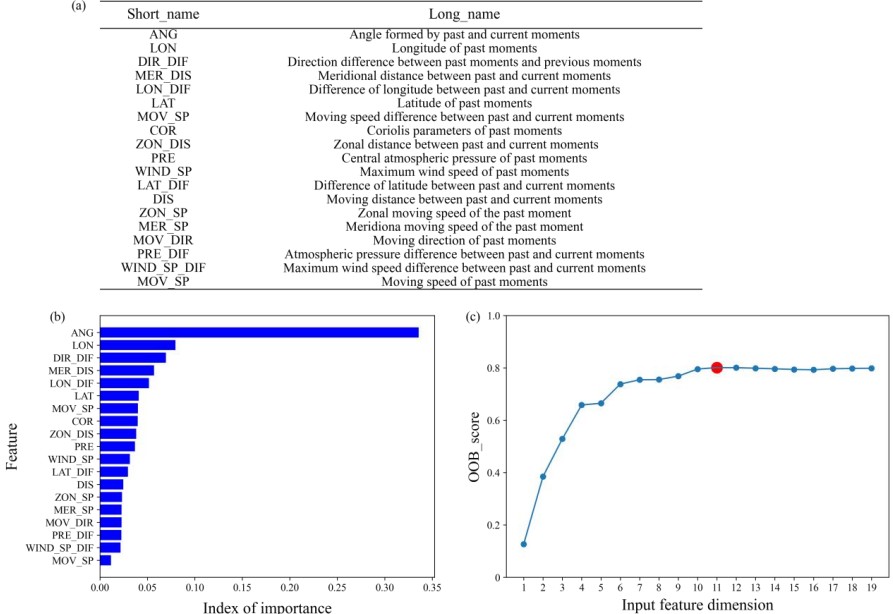


**Figure 1: (a) Table displaying the short and long names of features, (b) the importance index of features,**
**and (c) the OOB_score of different feature combinations based on the random forest (red dot indicates the**
**maximum value).**
**2.2.2 Recurrent Neural Network**
An RNN is an ANN characterized by architectural features intentionally designed to preserve historical
information, showing a remarkable ability to process sequential data. RNNs can process sequences of
any length using neurons with self-feedback, which has been successfully applied in many fields, such
as speech recognition (Graves et al., 2013), stock market predictions (Bathla, 2020), and trajectory
predictions (Wang and Fu, 2020). However, simple RNNs have difficulty in dealing with the long-term
dependence of the sequence; when the sequence length exceeds a certain threshold, the information
may disappear during the transmission process, resulting in large deviations in prediction accuracy. The
LSTM network proposed by Hochreiter and Schmidhuber (1997) can avoid the gradient disappearance
and explosion phenomena that occur in the standard RNN. While GRU (Cho et al., 2014) is an
improved and optimized neural network based on LSTM, it has a faster convergence speed and
maintains accuracy levels close to those of LSTM.

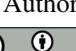



An LSTM layer consists of multiple recurrently connected blocks called memory blocks. Each
block has three multiplication units: an input gate, an output gate, and a forget gate. These three gates
perform their duties in a manner similar to filters. The input gate updates the state of the memory cell,
the forget gate selects relevant information from the previous memory cell, and the output gate controls
the output of each LSTM unit. The value of the hidden layer $h_t$ (Eq. 16) depends not only on the input
variable $x_t$ at the current moment but also on the hidden layer at the previous moment $h_{t-1}$. The
forward propagation formula of LSTM is as follows:
$$i_t = \sigma(W_i \cdot [h_{t-1}, x_t]) \tag{11}$$

$$\tilde{c}_t = \tanh(W_c \cdot [h_{t-1}, x_t]) \tag{12}$$

$$f_t = \sigma(W_f \cdot [h_{t-1}, x_t]) \tag{13}$$

$$c_t = f_t \cdot c_{t-1} + i_t \cdot \tilde{c}_t \tag{14}$$

$$o_t = \sigma(W_i \cdot [h_{t-1}, x_t]) \tag{15}$$

$$h_t = o_t \cdot \tanh(c_t) \tag{16}$$

Where σ and tanh are the sigmoid function and hyperbolic tangent activation function, respectively; $i_t$,
$f_t$, and $O_t$ are the input gate (Eq. 11), forget gate (Eq. 13), and output gate (Eq. 15), respectively; and
$C_t$ (Eq. 14) represents the cell state, the sizes of which are equal to the hidden vectors $h_{t-1}$. The
weight matrices $W_c$, $W_f$, and $W_i$ constitute the internal parameters of the neural network optimized
by the back-propagation algorithm.
The principle of GRU is very similar to that of LSTM. GRU merges the forget gate and the input
gate in LSTM to become the update gate $z_t$ (Eq. 17), and the output gate is also named the reset gate $r_t$
(Eq. 18). Intuitively, the reset gate controls the information from the current moment to the memory
unit $\tilde{h}_t$ (Eq. 19), and the update gate determines the amount of previous memory information saved to
the current time.
$$z_t = \sigma(W_z \cdot [h_{t-1}, x_t]) \tag{17}$$

$$r_t = \sigma(W_r \cdot [h_{t-1}, x_t]) \tag{18}$$

$$\tilde{h}_t = \tanh(W_{\tilde{h}} \cdot [r_t \cdot h_{t-1}, x_t]) \tag{19}$$



$$h_t = (1 - z_t)h_{t-1} + z_t \cdot \tilde{h}_t \qquad (20)$$

**2.2.3 Convolutional Neural Network**
A CNN is a hierarchical structure divided into convolution, pooling, and fully connected layers. CNN
can extract features automatically by processing the input patterns and translating the same convolution
kernel from top to bottom and from left to right. The spatial relationship is fixed with the distribution of
neurons, and the local connection and weight sharing of neurons reduce the training complexity by
reducing the number of parameters.
Lecun et al. (1998) first used CNN for handwritten character recognition with average pooling and
the tanh activation function. Krizhevsky et al. (2012) proposed the AlexNet model in the ImageNet
competition, using the ReLU function instead of the traditional tanh function to introduce nonlinearity
and solve the gradient disappearance problem of the activation function when the network was
relatively deep, employing maximum pooling to avoid the blurring effect of average pooling. Ioffe and
Szegedy (2015) applied batch normalization to image classification models, which significantly
accelerated the training of deep networks, and batch normalization helped alleviate the problem of
gradient exploding or vanishing.

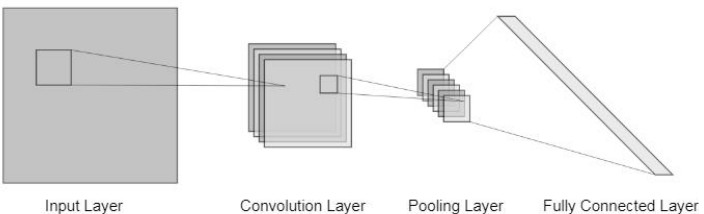


**Figure 2: Diagram showing the typical structure of a CNN.**
As shown in Fig. 2, all units in the same feature map share the same convolution kernel. The role of the
pooling layer is to improve the operating efficiency of the algorithm and expand the perception range
so that the subsequent convolution kernels can learn more global information. Typical operations
include the average and maximum pooling values. The fully connected layer is usually located at the
back end of the neural network, where all the features obtained from the hidden layer are expanded into
a vector connected to the neurons of the output layer.



### 3 Experiment

#### 3.1 Experimental design

Our goal is to predict the TC movements for the following 6–72 h using the trajectory data and surrounding environmental field from the previous 24 h. We explore TC movement in the Northwest Pacific from 1979 to 2021 and consider the longitudinal and latitudinal changes in the following 6–72 h as the quantitative prediction variables, with the center of the TC at the current time as the reference point. Since the maximum forecast hour is 72 and the input sequence time length is 24 h, TCs that persist for longer than 96 hours are removed. All samples obtained based on the sliding window of the input-prediction sequence length are divided into three groups in chronological order: training set (1979–2014), validation set (2015–2018), and test set (2019–2021). There are 36473 samples, of which 90% are trained, and the remaining 10% are validated; 49 TCs from 2019 to 2021 are used for testing, and the number of test samples is 2095.

#### 3.2 Model framework

Due to differences in the data sources, a new model must be developed to integrate the four information sources into the neural network using the Keras deep-learning framework. The specific model structure is shown in Fig. 3. For the 3D meteorological data, the data are superimposed on the geopotential height (pressure level), so the input data for the CNN consist of multiple two-dimensional matrices. The TimeDistributed layer is applied to a series of tensors in the processing of the time dimension. In addition, the CNN adopts a typical architecture with alternating convolution layers (Conv layers) and maximum pooling layers (Maxpool layers), converting two-dimensional data into one-dimensional vectors at the end of the CNN network and eventually creating a fully connected layer. All hidden layers are equipped with batch normalization, and this paper employs ReLU as the activation function.

For the two-dimensional trajectory data of the TCs, $x_i^j$ represents the input value of the $i^{th}$ feature at the $j^{th}$ timestamp, $i \in (1, n)$, $j \in (1, t)$, and they are input into GRU. The model is based on the Adam optimizer and trained with the RMSE between the forecast and the actual value as a loss function. Due to the different properties among the wind field, pressure field, SSTs, and past trajectory data, different





learning rates are required for the neural network. Therefore, the parameters of each branch in the
model can be trained with the same task, and then the branches can be fused into one network and
stitched with a fully connected layer; thereafter, the parameters can be adjusted slightly.

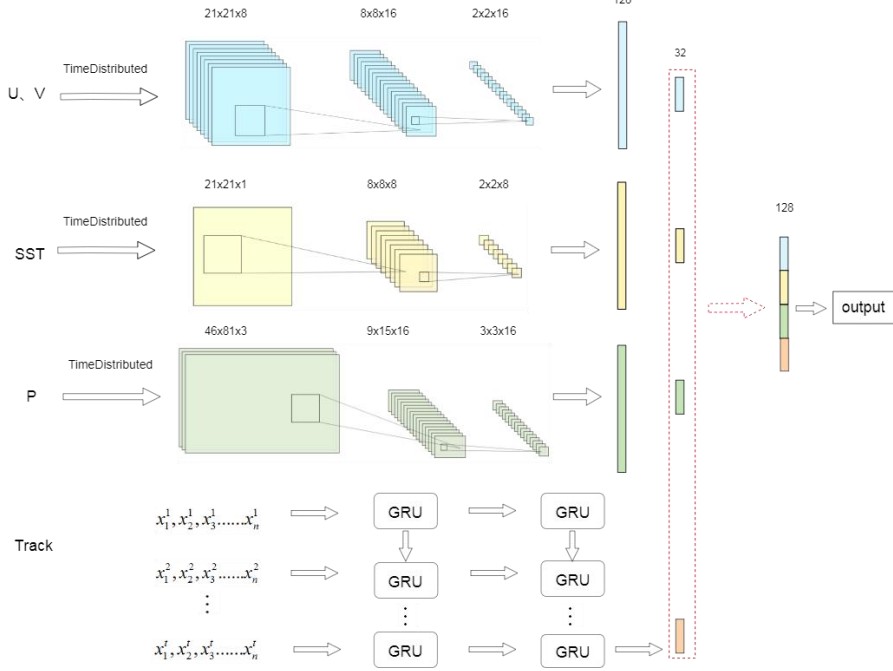


**Figure 3: The model framework and network structure of GRU_CNN.**
**3.3 Data normalization**
The differences in the dimensions of the input data will cause a variable with a larger value to have a
more significant impact on the model. Therefore, it is necessary to normalize the data before model
training to map the input data from 0 to 1. To eliminate the influence of different dimensions on the
model, the original value x is normalized as $x^{'}$:

$$x^{'} = \frac{x - x_{min}}{x_{max} - x_{min}} \qquad (21)$$

Where $x_{max}$ and $x_{min}$ are the maximum and minimum values of the variable x, respectively.




**3.4 Evaluation criteria**
RMSE is the root mean square error obtained by calculating the predicted value $P_i$ and the observed
value $O_i$. The formula is as follows:
$$RMSE = \sqrt{\frac{\sum_{i=1}^{n}(P_i - O_i)^2}{n}} \tag{22}$$

Since the latitude and longitude represent different spatial distances in kilometers, when comparing
other models, the distance error (Dis) is calculated in kilometers. The formula is as follows:
$$Dis = R \times arccos(\cos(Lat_{pred})\cos(Lat_{obs}) \times cos(Lon_{pred} - Lon_{pred})$$
$$+ \sin(Lat_{obs}) \times \sin(Lat_{pred})) \tag{23}$$

Where R is the radius of the earth, $Lat_{obs}$ and $Lon_{real}$ are the actual latitude and longitude, and
$Lat_{obs}$ and $Lon_{pred}$ are the predicted latitude and longitude.
**4 Results**
Three types of recurrent neural networks (RNN, LSTM, GRU) are used to train samples with eight
timestamps and 11 features selected by the random forest method, according to their importance; the
results of analyzing 49 TCs in 2019–2021 are then evaluated. We set the value of the batch size to 64
and the epoch to 100 and found that the model performed best when the number of neurons in the
hidden layer is set to 128; this was determined via experiments using different numbers of neurons in
the hidden layer. Early stopping is used to prevent overfitting. When the performance of the model in
the validation set begins to decline, training is stopped to avoid overfitting due to continued training.
Figure 4 shows the predicted latitudes and longitudes based on three RNNs in the following 6 h, 12 h,
18 h, 24 h, 48 h, and 72 h, and the results of all three networks within 24 h are all approximately
consistent with the real data. As the forecast time increases, the error accumulates.

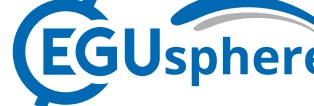

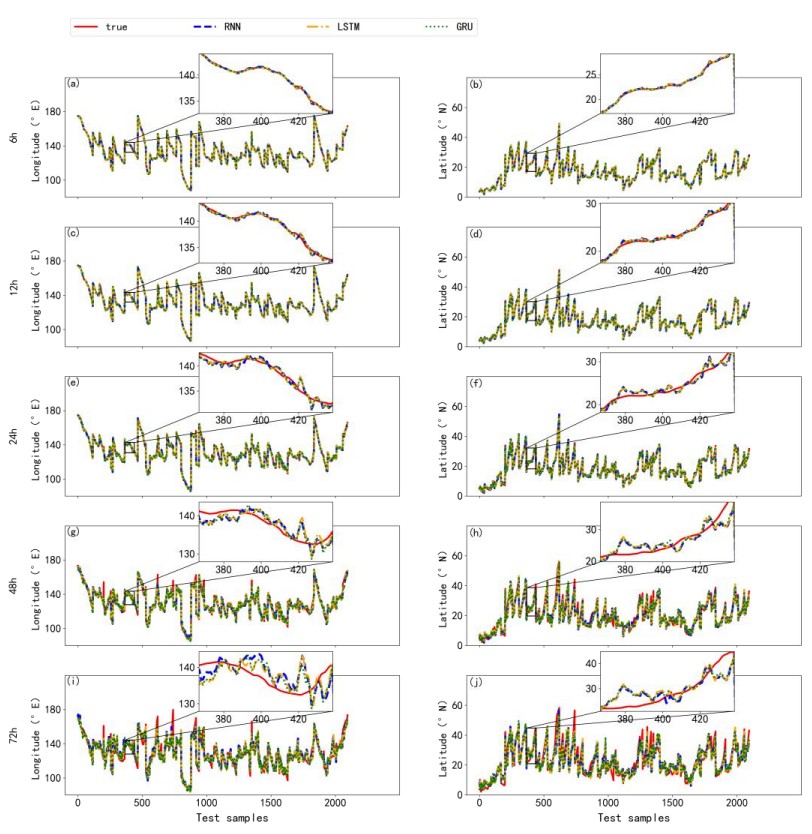


**Figure 4: Comparing the longitudes and latitudes of all test samples predicted by the three recurrent neural**

**networks (RNN, LSTM, GRU), including the longitude/latitude forecasts at (a–b) 6 h, (c–d) 12 h, (e–f) 24 h,**

**(g–h) 48 h, and (i–j) 72 h.**




**Table 1 Model performance evaluation (RMSE) for RNN, LSTM, and GRU**

| Forecast Hour | | Lat | | | | | | Lon | | | | | |
|---|---|---|---|---|---|---|---|---|---|---|---|---|---|
| | | 6 h | 12 h | 18 h | 24 h | 48 h | 72 h | 6 h | 12 h | 18 h | 24 h | 48 h | 72 h |
| Training Sets | RNN | 0.146 | 0.368 | 0.624 | 0.912 | 2.232 | 4.069 | 0.153 | 0.412 | 0.735 | 1.118 | 3.181 | 6.236 |
| | LSTM | 0.126 | 0.335 | 0.584 | 0.867 | 2.34 | 3.986 | 0.149 | 0.391 | 0.703 | 1.077 | 3.232 | 6.172 |
| | GRU | **0.112** | **0.312** | **0.555** | **0.83** | **2.282** | **3.883** | **0.134** | **0.376** | **0.681** | **1.041** | **3.152** | **6.031** |
| Validation Sets | RNN | 0.171 | 0.441 | 0.76 | 1.116 | 2.723 | 4.493 | 0.182 | 0.483 | 0.865 | 1.337 | 3.739 | 6.715 |
| | LSTM | 0.157 | 0.428 | 0.751 | 1.103 | **2.675** | **4.349** | 0.183 | **0.480** | **0.855** | 1.305 | 3.69 | 6.761 |
| | GRU | **0.157** | **0.42** | **0.735** | **1.086** | 2.699 | 4.434 | **0.179** | 0.484 | 0.868 | **1.33** | **3.632** | **6.608** |
| Test Sets | RNN | 0.166 | 0.411 | 0.685 | 0.988 | 2.481 | 4.207 | 0.176 | 0.461 | 0.797 | 1.178 | 3.534 | 6.157 |
| | LSTM | **0.149** | 0.389 | 0.661 | 0.965 | 2.473 | 4.154 | 0.169 | **0.456** | 0.812 | 1.215 | 3.346 | 5.989 |
| | GRU | **0.149** | **0.387** | **0.653** | **0.951** | **2.446** | **4.143** | **0.167** | 0.457 | **0.8** | **1.185** | **3.325** | **5.969** |


The performance evaluation of the three RNN models is displayed in Table 1 by calculating the RMSE
values between the predicted longitude (latitude) and the actual longitude (latitude), including training,
validation, and test sets; the best results are highlighted in bold font. It is clear that the GRU-based and
LSTM-based models significantly outperformed the RNN-based model, which suggests that the RNN
is inferior in handling the problem of long-term dependence. GRU is a variant of LSTM that combines
the forget and input gates in LSTM into an update gate and also merges the cell and hidden states.
Hence, the parameter amounts of GRU are less than those of LSTM, which results in the overall
training speed of GRU being faster than that of LSTM. GRU is theoretically similar to LSTM and can
achieve the same accuracy as LSTM (or even better), so the results of GRU and LSTM are close and
their RMSE values are much lower than that of RNN. GRU achieves the best performance in all
forecast hours, with the smallest RMSE in the test set. Therefore we use GRU as a part of the fusion
network model called GRU_CNN, adding meteorological environment data processed with CNN.
**Table 2 Comparison of the average absolute distance errors (km) predicted by multiple deep-learning**
**models**

| | 6 h | 12 h | 18 h | 24 h | 48 h | 72 h |
|---|---|---|---|---|---|---|
| CLIPER(Demaraia,1992) | — | — | — | 213 | 442 | 659 |
| BP | 23.86 | 59.58 | 101.01 | 146.91 | 377.64 | 634.42 |
| RNN | 21.43 | 55.46 | 94.59 | 138.09 | 373.12 | 625.17 |
| LSTM | 19.65 | 52.38 | 91.76 | 136.05 | 360.32 | 614.76 |
| GRU | 19.51 | 52.6 | 91.21 | 134.73 | 357.25 | 607.44 |
| NMSTN(Huang, 2022) | 27.52 | 59.09 | — | 139.18 | 336.16 | 544.16 |
| GRU_CNN | 17.22 | 43.9 | 72.74 | 106.16 | 281.52 | 502.71 |




Table 2 compares the results between GRU_CNN and various deep-learning models, showing the
forecast results in the form of the mean absolute distance error. It is evident that GRU_CNN presents
an absolute advantage in long-term forecasting. Both LSTM and GRU retain important features
through various gate functions, which ensures that they will not be lost during long-term propagation.
They can better predict the medium and long-term tracks of the TCs, compared with standard RNNs
and two traditional methods named CLIPER and BP. The GRU_CNN is more accurate than the models
without CNN. The average distance errors at 6 h, 24 h, 48 h, and 72 h are 17.22 km, 106.16 km, 281.52
km, and 502.71 km, respectively. The error is also reduced compared with the NMSTN method
proposed by Huang et al. (2022). In addition, although there is a big difference between the long-term
forecast and the numerical prediction results, the average distance prediction results are better than the
results provided by the Central Meteorological Observatory (CMO) in the short-term forecasts,
including the 6 h (27.57 km) and 12 h (59.09 km) forecasts.

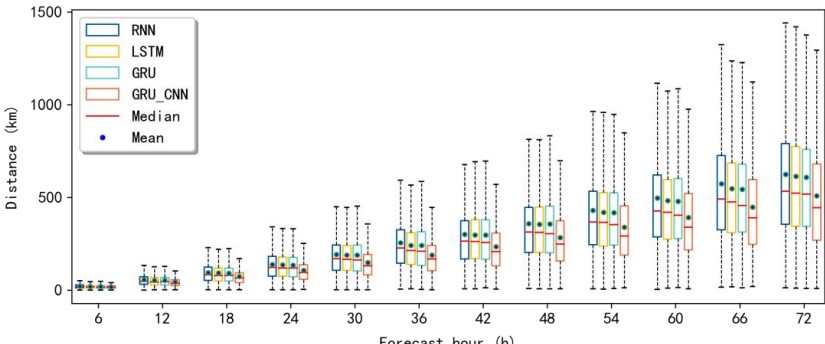


**Figure 5: The absolute average distance boxplot of the three kinds of recurrent neural networks (RNN,**
**LSTM, GRU) and the method in this paper (GRU_CNN) creating 6–72 h forecasts (interval 6 h).**
As shown in Fig. 5, the maximum distance errors predicted by the three RNNs at 48 h and 72 h are
over 500 km and 1000 km, respectively. Only considering the trajectory characteristics of the TCs in
the RNN while ignoring the external atmospheric environmental characteristics will cause instability in
the prediction of the TC tracks. The errors of the maximum and average values predicted by the
GRU_CNN model are both significantly reduced. To illustrate GRU_CNN more comprehensively and
intuitively, Fig. 6 shows a scatter plot of the predicted and actual values. The distance between the data
points and the diagonal line represents the prediction error. The higher the wind speed, the stronger the





intensity of the TCs, and the closer the predicted value is to the actual value. In addition, with the
increase in the forecast time, in high latitude and longitude forecasts when the TC is moving towards
the northwest, the predicted value is often lower than the actual value.

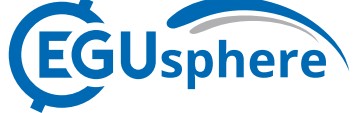

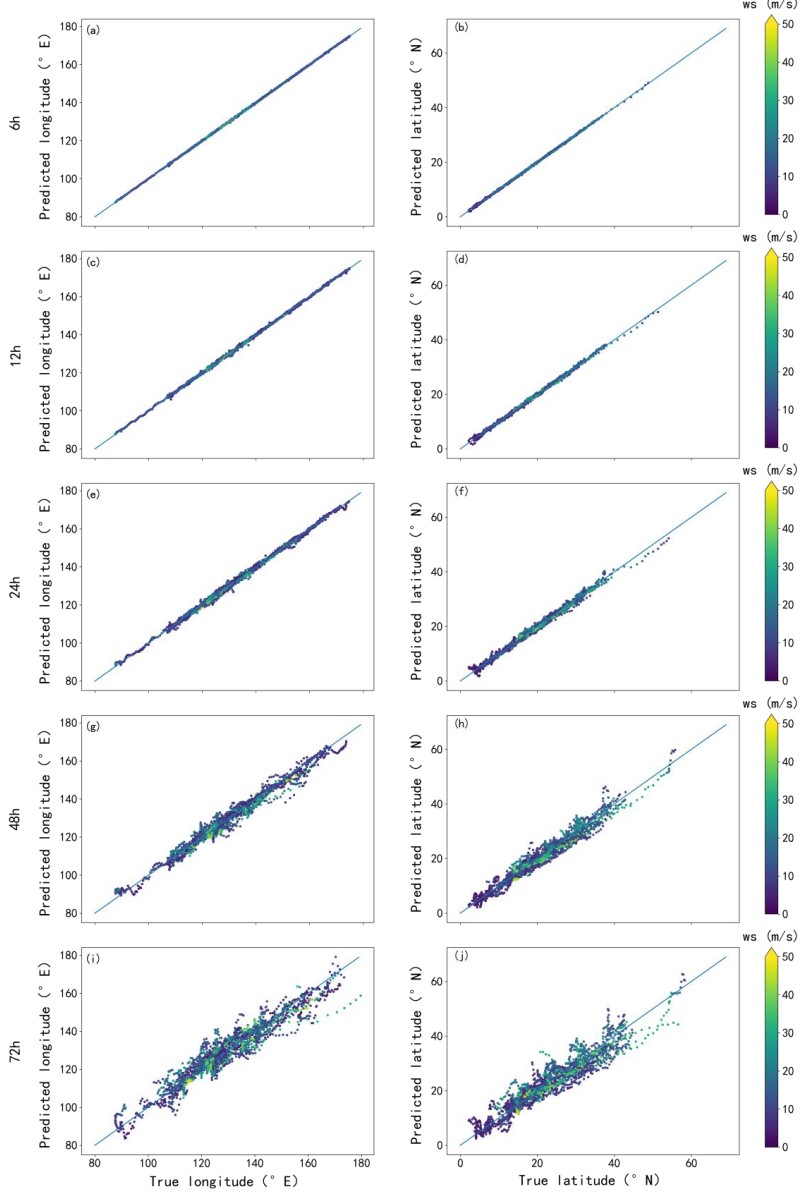


**Figure 6: Scatter plot distributions of latitude predictions. The color bar represents the maximum wind**


**speed, including the longitude and latitude forecasts at (a–b) 6 h, (c–d) 12 h, (e–f) 24 h, (g–h) 48 h, and (i–j)**


**72 h.**





Data from three environmental fields are used in this paper: SST, geopotential height (pressure),
and wind field (*u*- and *v*-component) data. Different environmental input variables show different
effects in the model (Table 3). GRU+SST (pressure, UV) represents only the combination of the
trajectory characteristics and SST (geopotential height, wind field), while GRU+CNN is the result of
the fusion of the three. The results in Table 3 indicate that GRU+UV performed best, followed by
GRU+pressure and then GRU+SST, indicating that the steering flow plays a dominant role in TC
forecasting, especially in the short-term < 24-h forecast. The forecasting results of adding only the
steering flow are close to those of GRU_CNN, while the results at 48 h and 72 h illustrate that the
influence of the SST and geopotential height on the long-term TC forecast track gradually increases.
**Table 3 Comparison of trajectory data combining different environmental features. RMSE is the root mean**
**square error of latitude and longitude, and the distance is the average absolute distance error (km).**

|      | GRU+SST | | GRU+Pressure | | GRU+UV | | GRU+CNN | |
|------|------|---------------|------|---------------|------|---------------|------|---------------|
|      | RMSE | Distance (km) | RMSE | Distance (km) | RMSE | Distance (km) | RMSE | Distance (km) |
| 6 h  | 0.154 | 19.35 | 0.132 | 16.15 | 0.137 | 16.94 | 0.138 | 17.22 |
| 12 h | 0.419 | 52.37 | 0.352 | 44.22 | 0.347 | 43.74 | 0.35 | 43.9 |
| 18 h | 0.739 | 92.25 | 0.598 | 76.16 | 0.575 | 72.81 | 0.575 | 72.74 |
| 24 h | 1.103 | 137.78 | 0.883 | 112.93 | 0.841 | 106.63 | 0.837 | 106.16 |
| 48 h | 2.858 | 358.25 | 2.462 | 306.03 | 2.379 | 302.76 | 2.248 | 281.52 |
| 72 h | 4.913 | 588.08 | 4.52 | 557.86 | 4.385 | 524.88 | 4.146 | 502.71 |


To better show the model forecast of GRU_CNN, Figures 7–9 present the observed and forecast
tracks at 6 h and 24 h of TCs FAXAI, MITAG, and IN-FA, respectively, and the forecast tracks of
other TCs in the test set are presented in Supplementary Fig. S1-51. The blue lines represent the
observed tracks, while the red and yellow lines indicate the 6-h and 24-h forecast tracks. In general, it
is particularly hard to forecast unexpected turns in the TC track. The three TCs shown all exhibit a
sudden northward or northwestern turn in the TC track. For the 6-h forecast, the predicted path is
approximately consistent with the actual track, while the 24-h forecast has some deviations. The
average distance predicted near the northwest turn of FAXAI is 91.35 km; the error for MITAG's first
turn to the north is 127.02 km, and the error for the second turn to the northwest is 121.91 km. The two
average errors in the track forecast for In-fa are 84.27 km and 82.37 km. It can be seen that there is no
significant deviation in the forecast around the steering point, but, for some abnormal track changes,



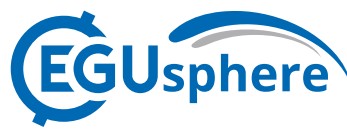

such as crossing back over the same location, samples with more significant errors will be generated,
reducing the overall average absolute distance error.

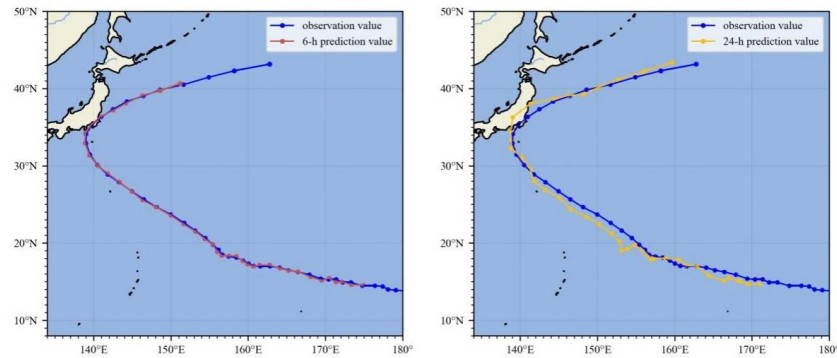


**Figure 7: Forecast tracks of Tropical Cyclone FAXAI (1915) (left: 6 h, right: 24 h).**

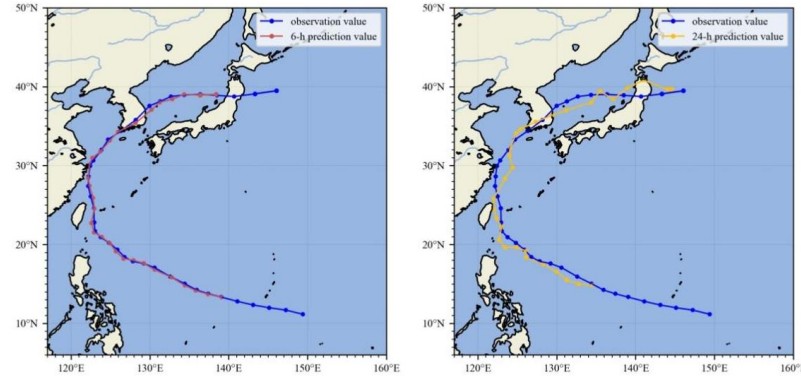


**Figure 8: Forecast tracks of Tropical Cyclone MITAG (1918) (left: 6 h, right: 24 h).**



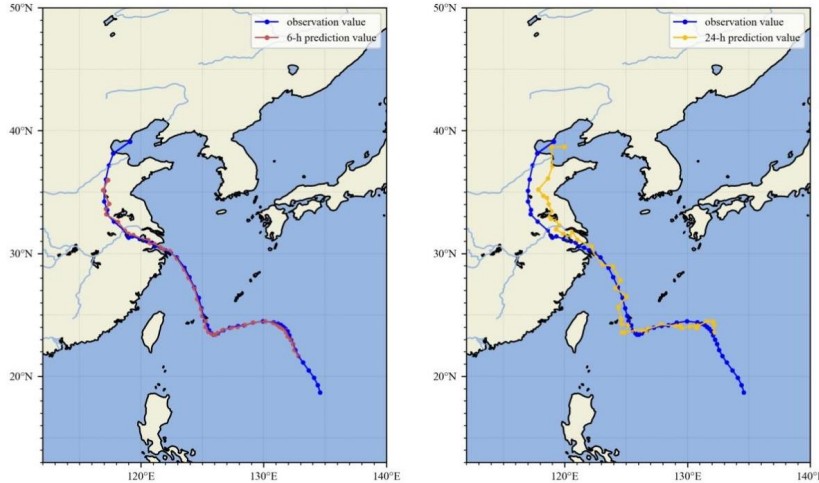


**Figure 9: Forecast tracks of Tropical Cyclone IN-FA (2106) (left: 6 h, right: 24 h).**
**5 Conclusion**
The past 24-h TC trajectory and meteorological field data have been used to forecast TC tracks in the
Northwest Pacific from hours 6–72. First, in order to eliminate data redundancy and reduce the
complexity of the prediction model, the random forest algorithm was used for feature extraction of the
two-dimensional movement data. Second, three kinds of recurrent neural networks (RNN, LSTM,
GRU) were used to evaluate and compare the models based on the input of trajectory features, and it
was concluded that GRU performed relatively better in predicting TC tracks. Eventually, we combined
GRU with CNN by adding the pre-processed meteorological environmental data around the cyclones
(removing the vortex to obtain the steering flow); the CNN models the selected meteorological
variables and extracts features, while GRU processes trajectory sequences. GRU_CNN has better
prediction results than traditional single deep-learning methods do.
When a new TC generates in the ocean, the GRU_CNN model can quickly provide the forecast
track within seconds. Short-term predictions within 12 h of initialization can provide better results than
CMO can, and the average distance errors of the forecasts at 6 h and 12 h are 17.22 km and 43.9 km.
When the forecast goes beyond 24 h, the model's accuracy declines. The historical steering flow of





cyclones has a significant effect on improving the accuracy of short-term forecasting, while, in
long-term forecasting, the SST and geopotential height will have a particular impact, which is regarded
as a crucial way to expand and improve the application of deep-learning models in TC track forecasting,
In addition, the model can accurately predict TCs that suddenly turn to the north or northwest, but there
will be a considerable distance error for abnormal trajectories, possibly due to a lack of synoptic
analysis in our study.

Cyclone prediction has been a challenge in weather forecasting for a long time. With future

scientific and technological advances, it is becoming increasingly convenient to obtain meteorological
data, and the database has gradually expanded. At the same time, deep-learning models are flexible and
can easily be expanded upon. In the future, more data can be integrated, and more valuable features can
be extracted to improve the prediction accuracy of the deep-learning model. In addition, model
predictor variables will be considered in future work, the inclusion of which can predict more useful
information, such as cyclone intensity, rainfall, and wind speed.
*Code availability.* The code and model are available as a free access repository on GitHub at
https://github.com/Hush980/TCs_DL_code.
*Data availability.* IBTrACS that we used in this study is publicly available. It can be down-loaded at
https://www.ncei.noaa.gov/data/international-best-track-archive-for-Climate-stewardship-ibtracs.
ERA5 data can be obtained from Copernicus Climate Data Store (https://cds.climate.copernicus.eu).
*Author contributions.* Liang Wang wrote the paper and conducted most of the code implementation and
data analysis. Bincheng Wan designed the research framework. Shaohui Zhou provided the code and
revised the paper. Haofei Sun was involved in data collation and Zhiqiu Gao was responsible for
supervision.
*Competing interests.* The authors declare that they have no known competing financial interests or
personal relationships that could have appeared to influence the work reported in this paper.



*Acknowledgments.* This study was supported by the National Key Research and Development Program
of the Ministry of Science and Technology of China (2018YFC1506405), and by the National Natural
Science Foundation of China (Grants 42175082 and 42222503).

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
