# Peer review of "Forecasting tropical cyclone tracks in the Northwest"

_EGUsphere, 2022_

## Author Comment (AC3)

We would like to thank the reviewer for his careful reading of the manuscript and for their comments and suggestions that helped improve this manuscript. All comments have been addressed and a point-by-point answer is provided in the following (in blue after the corresponding comment).

General comments:

The authors spend a lot of space explaining the technical details of the standard machine learning and deep learning models (random forests, RNN, CNN). These can be shortened, or refer the readers to the detailed background references. In addition, the equations for the GRU and LSTM cells are hard to follow, thus they can be complemented with diagrams showing the flow of data in these cells.

**Reply:** Thank you for your valuable advice. We reorganized the structure of sections 2 and 3 in the revised manuscript, merging the random forests method and de-vortexing method into data preprocessing, and merging RNNs and CNNs into the model framework of section 3.2. The standard background part of the three methods presented in the manuscript (Random Forest/RNNs/CNNs), was deleted. In addition, the relevant diagrams of LSTM and GRU are added to the revised manuscript so that readers can better understand it.

In the model framework section, I find it hard to understand the network architecture that the authors used in this work.

I think it would be benefit to include a table detailing the network architecture.

In addition, in figure 3, I think the description of the figure could be revised to include more details such as: CNN kernel size, what the solid white arrows mean, what the dashed red arrow means, etc.

The authors do not mention the architecture of the RNN, LSTM, GRU that they used in this work. I think it would improve the clarity if they were included here.

**Reply:** Thank you for your valuable advice. We added an explanation of the arrows in Figure 3. Among them, the gray filling arrow represents the TimeDistributed layer that is applied to a series of tensors in the processing of the time dimension. The black solid arrow means the multidimensional tensor into a one-dimensional vector. The dashed black arrow represents the fully connected layer in the network framework, and the dashed red arrow means the merging of multiple vectors into one vector. The figure is revised as follows:

[Figure]

**Figure 3: The model framework and network structure of GRU_CNN.**

In addition, In order to show the network framework more clearly and facilitate readers reading and understanding, we added a table to list the input and output size of each layer in the network framework, including convolution kernel size, stride, and channel number.
The 2D-CNN-based encoder architecture of the pressure field branch

**Table 1 Each layer architecture of the GRU_CNN**

| Layers | Kernel Size | Stride | Channel | Input Size | Output Size |
|---|---|---|---|---|---|
| Conv_uv | 7×7 | 2 | 8 | 21×21 | 8×8 |
| MaxPool_uv | 4×4 | 4 | 16 | 8×8 | 2×2 |
| Flatten _uv | - | - | 16 | 2×2 | 64 |
| Dense_uv_1 | - | - | - | 64 | 128 |
| Dense_uv_2 | - | - | - | 128 | 32 |
| Conv_sst | 7×7 | 2 | 1 | 21×21 | 8×8 |
| MaxPool_sst | 4×4 | 4 | 8 | 8×8 | 2×2 |
| Flatten_sst | - | - | 8 | 2×2 | 32 |
| Dense_sst_1 | - | - | - | 32 | 128 |
| Dense_sst_2 | - | - | - | 128 | 32 |
| Conv_p | 14×25 | 4 | 3 | 46×81 | 9×15 |
| MaxPool_p | 5x11 | 4 | 16 | 9×15 | 2×2 |
| Flatten _p | - | - | 16 | 2×2 | 64 |
| Dense_p_1 | - | - | - | 64 | 128 |
| Dense_p_2 | - | - | - | 128 | 32 |
| GRU_1 | - | - | - | 8×11 | 8×128 |

| | | | | | |
|---|---|---|---|---|---|
| GRU_2 | - | - | - | 8×128 | 128 |
| Dense_GRU | - | - | - | 128 | 32 |
| Concat_layer | - | - | - | - | 128 |

Line 323-327 introduces some detailed descriptions of the three recurrent neural network frameworks. Firstly, RNN, LSTM, and GRU are all recurrent neural networks with similar structures and the parameters of the three networks are the same. Secondly, their architectures are actually included in the overall framework, which is a part of our proposed model GRU_CNN, so it is not highlighted. In the case of only inputting trajectory features, these three networks are used to compare which result is better, and then put it into our fusion model.

In the discussion of table 3 (L384-L392), the authors claim that the influence of SST and geopotential height gradually increases at long-term forecasts. Can the authors provide more explanation of why this is the case?

**Reply:** This conclusion is based on the statistically average value in table 3, which can not represent each tropical cyclone and can be regarded as the results of the whole. It is shown that as the forecast time increases, the proportion of the steering flow to the predicted value gradually weakens, so the sea surface temperature and geopotential height increase accordingly. However, we have not found relevant literature to explain this phenomenon. Meteorologically, sea surface temperature will drive the TCs to the warm sea surface, so it will not affect the movement of the TCs in a short time. The geopotential height represents the weather system at high and low altitudes, and it will affect the movement of TCs for a long time. When there is a subtropical anticyclone staying in the north of the TC, it will cause the cyclone to stagnate or move slowly, which involves the analysis of weather patterns in meteorology, It is very interesting, but beyond the scope of this manuscript.

Since the authors compare the performance of GRU_CNN with other methods: FAXAI, MITAG, and IN-FA in figures 7-9, I think it would be more convincing if the authors can also provide detailed comparison between these models like in the table 3.

**Reply:** Here, FAXAI, MITAG, and IN-FA are all the TCs' names, not the model method. We selected three of the 54 typhoons in the test sets and analyzed them. Figure 7-9 shows their actual and predicted paths. They all have the common characteristics of track turning. The other 51 TCs' forecast paths are shown in the supplementary document.

Specific comments:

L79: missing a space between a reference and the word "applied"
**Reply:** Suggestion adopted.

Figure 4: this figure could instead show the difference between the predicted longitudes/latitudes with the observed longitudes/latitudes to improve clarity and readability.
All figures' texts and labels can be a bit bigger to improve readability.
**Reply:** Suggestion adopted.

L397: what are these methods: FAXAI, MITAG, IN-FA? Can you provide a short description and references for these methods?

**Reply:** This question has been answered above in the fourth General comment.

---

## Author Response (AR1)

Dear Dr. Kieu

On November 19 last year, The manuscript titled as "Forecasting tropical cyclone tracks in the Northwest Pacific based on a deep-learning model" was accepted for EGUsphere preprint posting and discussion. We thank you very much for your efforts in evaluating our submission and thank Reviewers 1 and Reviewer 2 for their careful reading of the manuscript and for the comments that improved the quality of the manuscript.

All comments have been addressed and a point-by-point answer is provided in the following (in blue after the corresponding comment). In addition, the red line number and the number of the table or figure represent themselves in the revised manuscript (the blue line number and the number of the table or figure correspond to them in the initial manuscript ).

I hereby resubmit the revised paper to you to be considered for publication in the Journal of Geoscientific Model Development.

I confirm that all authors listed on the manuscript concur with submission in its revised form. Should you have any remaining questions, I will be happy to address them.

Sincerely yours,

Liang Wang with a grateful heart
Address: Institute of Atmospheric Physics, Chinese Academy of Science
Beijing, China
Email: wangjing202@mails.ucas.ac.cn
February 12 2023

**Reviewer #1**

Major comments:

1. The manuscript is lengthy at 28 pages, but nearly half are devoted to introduction and methods that use a lot of space to describe standard procedures in machine learning. For example, section 2.2.1 goes into great detail about OOB samples. A section in feature selection using random forest is necessary (Figure 1 is very good) but the background is very standard, and can be shortened and refer the reader to appropriate background references. Section 2.2.2, 2.2.3 include a lot of background in RNNs and CNNs that can also be shortened as it is standard procedure in machine learning and not specific to forecasting cyclone tracks. Same for 3.3, 3.4 in normalization and evaluation criteria which do not need to be in separate sections; overall, the sections before the results can be reorganized for conciseness and avoid replicating a lot of existing background literature on the topic.

**Reply:** Thank you for your valuable advice. We reorganized the structure of sections 2 and 3 in the revised manuscript, renaming "2 Data and methods" to "2 Data and data preprocessing", merging 2.2.1 devortexing method and 2.2.2 random forests method into 2.2 data preprocessing, and merging RNNs and CNNs into the model framework of section 3.2. The standard background part of the three methods presented in the manuscript (Random Forest/RNNs/CNNs), was deleted and replaced with references, besides, we deleted sections 3.3 and 3.4.

2. I would like to suggest authors to be more careful in the introduction in regards to the strengths and limitations of NWP, statistical models, and deep learning, avoiding potential biases towards methods that are not deep learning. For example, line 49-50 says that NWP models have "limitations in methods" requiring "numerous calculations". Framing of these limitations is needed. Is computational performance of these numerous calculations unacceptable? The work presented in this manuscript is of course efficient, giving results in seconds. But how long do NWP models take? The authors then mention "accurate mathematical descriptions of physical atmospheric mechanisms". NWP models aren't always exact and can involve many approximations and parameterizations. The authors may be trying to convey that NWP models require description of physical processes versus machine learning methods that learn from data. But that is not a limitation, it would be a property of different approaches to modeling.

**Reply:** Thank you for your valuable advice. We agree with the reviewer that these statements may be potentially biased. Lines 49-50 (Lines 49-50): "there are limitations in methods requiring numerous calculations, accurate mathematical descriptions of physical atmospheric mechanisms, and precise initial conditions" was changed to "there are limitations in methods relying on high-performance computers and requiring precise initial conditions."

However, due to the computer performances, mode resolutions, and sizes of the selected areas being different, it is difficult to determine the running time of numerical model prediction.

The authors also criticize statistical models, saying "manual feature selection is unable to produce accurate predictions". The inaccuracy would need to be characterized (and cited where appropriate) in order to reach this conclusion.

**Reply:** Lines 60-61 (Lines 61-62): We added the "CLP5 had the largest Mean absolute error(MAE)

of all models for TCs occurring from the Eastern Pacific and North Atlantic (Boussioux et al., 2022). " and "Li-Min et al. (2009) used the BP neural network to predict that the average distance error of the 6h movement track of six typhoons in 2005 improved by 36.9km, compared with CLIPER." after "manual feature selection is unable to produce accurate predictions." .

In the following paragraph about deep learning, authors give very specific accuracies (e.g., L83, L85, L91) and strengths of this method. For completeness and fair comparison, I suggest authors also give conventional methods similar statistics and strengths, and avoid vague, uncited description of limitations.

**Reply:** Lines 83-90 (Lines 80-87): "Gao et al. (2018) used long short-term memory (LSTM) to predict typhoon tracks in the Northwest Pacific Ocean; the ratio of the cyclone training set and test set was set at 8:2, and the 24-h prediction error could reach 105 km. Alemany et al. (2018) proposed an RNN based on a grid system to predict hurricanes in the Atlantic, potentially improving the 6-h prediction accuracy with a root mean square error (RMSE) of 0.11 for the test set. Kim et al. (2018) performed a TC identification task based on ConvLSTM to train WRF-simulated data, and the results are significantly better than those of a convolutional neural network (CNN)." introduce RNNs (one of the deep learning methods) applied to TC track forecasts that occurs in different sea basins and different years, resulting in different training sets and test sets. Another deep learning method, CNNs, is also introduced in Lines 93-95 (Lines 89-91): " Giffard-Roisin et al. (2020) combined historical trajectory data with wind field reanalysis data as input to a CNN and predicted Atlantic hurricane tracks since 1979, with an average error of 32.9130 km for 24-h predictions." They are not comparable. These examples are to illustrate the feasibility of the application of RNNs in TC track forecasts. In addition, the author only gives 6h prediction results in the article. It is difficult to unify the standard.

3. Text in figures is at times hard to read because of the small font size. Please also make the fonts consistent, e.g., Arial. I suggest going through the futures to ensure consistency in presentation and that all figures are clear and readable.

**Reply:** Thank you for your valuable advice. After careful examination, we adjusted the size of the text in the figures and unified the fonts for Times New Roman.

Specific comments:

1. Abstract, L29: please include the average distance errors of the CMO forecast results as well for comparison.

**Reply:** Line 29 (Line 29): "(27.57km and 59.09km)" has been added in the revised manuscript.

2.L82 uses 24-h prediction distance error for LSTM, then L85 uses 6-h RMSE, L91 uses 6-h distance error. If possible, please be more consistent in error metrics.

**Reply:** This question has been answered above in the third paragraph of the second major comment.

3. L86: authors say Kim et al. (2018) are "significantly better" than those of a CNN. How much?

**Reply:** Lines 89-90 (Line 86): "and the results are significantly better than those of a

convolutional neural network (CNN)." was changed to "and the results show the average precision of the forecast was improved by 78.99% than those of a convolutional neural network (CNN)."

4. L97: please define MLP, first time the acronym has appeared in the manuscript.
**Reply:** Suggestion adopted. Line 100 (Line 97): MLP is modified to the full name "Multi-Layer Perceptron".

5.L101: "Previous studies have shown…". Which previous studies? Please provide references.
**Reply:** Actually, what we want to express is the above studies. Line 104 (Line 101): we have modified it in the manuscript.

6. L102: "Still, most of them have neglected to describe and analyze the meteorological factors that affect the movement of TCs, ignoring valuable features." Which studies? Did this neglect of meteorological factors significantly affect performance, compared to studies that have considered these factors? Please also give examples of these "valuable features".
**Reply:** Thank you for your valuable advice. Lines 108-110 (Line 102): We added "The 6-hour average distance error between predicted and real location by the fusion network (wind+track) is 32.9 km, while the network prediction results without adding wind variables are 35km (Giffard-Roisin et al., 2018), which indicates that the addition of meteorological field variables can effectively improve the prediction accuracy."

7. L126-127: Do you mean that the Coriolis parameter is included in the predictors?
**Reply:** No, the Coriolis parameter of the TC is included in the input variable. In order to avoid ambiguous statements, Lines 130-131 (Lines 126-127): "In addition, the Coriolis parameter corresponding to the latitude of the past 24 h influences the geostrophic deflection force on the TCs." was changed to "The Coriolis parameters corresponding to the latitude of the TCs in the past 24 hours are also included."

8. L128-133 describes a TC bias to northwest; I am having trouble following the reason for this paragraph. Is this the reason for the geographically asymmetrical data selection in L142-147 (3)? If yes, then why is this not done for (1) & (2)?
**Reply:** There is a formatting error at the end of this paragraph that should be merged with the next paragraph.
I deleted this sentence "Because they are influenced by the earth's rotation, TCs will be biased to the northwest (Kitade, 1981)." and added it before "The Coriolis parameters corresponding to the latitude of the TCs in the past 24 hours are also included." in Lines 131-132 (Line 126) .
Lines 134-141 (Lines 128-137): "Both observational and theoretical studies have shown that TC movement is closely related to large-scale airflow fields (Holland, 1983), and TC movement is mainly affected by the steering flow (Brand et al., 1981; Chan, 1984). Interactions among weather systems, the subtropical anticyclone, Westerlies, and the Tibetan High will also affect the movement of cyclones (George and Gray, 1976; Chan et al., 1980). The geopotential heights of 300 hpa, 500 hpa, and 700 hpa are selected as the locations for the high, middle, and low-level circulation data, respectively. In addition, the underlying surface conditions must be considered, and, in the case of a weak guidance environment, TCs tend to move toward warmer sea-surface

temperatures (Sun et al., 2017; Katsube and Inatsu, 2016)." describes several meteorological factors affecting the TCs movement including the steering flow, sea surface temperature, and weather systems, corresponding to UV, SST and HGT described in the next content. First of all, I refer to a Chinese paper on the selection of the meteorological field variable division area size, and it is symmetrical in the zonal direction but asymmetrical in the meridian direction. The main reason is that TCs tend to move north, so they are mostly affected by the weather system in the north, especially the subtropical anticyclone.

9.  L143, L145 "10 degree radius". Do you mean extended by a 10 degree distance in each direction, since a 21x21 grid is formed?
**Reply:** Line 147, Line 150 (Line 143, Line 145): Here is my misstatement. The "radius of 10 degrees" should be changed to "centered the typhoon, extend 10 degrees outward in the zonal and meridian direction respectively, and form a square matrix with a 21x21 grid.

10.  Figure 4: I suggest also adding the RMSE for the test set for the three recurrent neural networks inset in the figures for ease of comparison.
**Reply:** The RMSE for the test sets for the three recurrent neural networks have been listed in Table 2 (Table 1). In addition, the comparison of the three recurrent neural networks is not the focus of this manuscript, and Figure 4 does not show a good contrast relationship, thus it is deleted after careful consideration.

11.  L414, Figure 9 legend: 2106 -> 2016.
**Reply:** The numbers in parentheses originally indicate the numbers of the TCs. After we review several works of literature, these numbers should be changed to years.

12.  L416-417: add "using deep learning methods" at the end of the opening sentence.
**Reply:** Suggestion adopted.

**reference**
Boussioux, L., Zeng, C., Guenais, T., and Bertsimas, D.: Hurricane Forecasting: A Novel Multimodal Machine Learning Framework, Weather and Forecasting, https://arxiv.org/abs/2011.06125, 2022.
Giffard-Roisin, S., Yang, M., Charpiat, G., Kégl, B., and Monteleoni, C.: Fused Deep Learning for Hurricane Track Forecast from Reanalysis Data, Climate Informatics Workshop Proceedings 2018, Boulder, United States, 2018-09-19, https://hal.science/hal-01851001, 2018.
Li-min, S., Gang, F. U., Xiang-chun, C., and Jian, Z.: Application of BP neural network to forecasting typhoon tracks, Journal of Natural Disasters, 18, 104-111, 2009.

**Reviewer #2**

General comments:

The authors spend a lot of space explaining the technical details of the standard machine learning and deep learning models (random forests, RNN, CNN). These can be shortened, or refer the readers to the detailed background references. In addition, the equations for the GRU and LSTM cells are hard to follow, thus they can be complemented with diagrams showing the flow of data in these cells.

**Reply:** Thank you for your valuable advice. We reorganized the structure of sections 2 and 3 in the revised manuscript, merging the random forests method and devortexing method into data preprocessing, and merging RNNs and CNNs into the model framework of section 3.2. The standard background part of the three methods presented in the manuscript (Random Forest/RNNs/CNNs), was shortened. In addition, the introduction of formulas for LSTM and GRU is a standard procedure in machine learning and not specific to forecasting cyclone tracks, therefore this part is deleted and replaced with background references.

In the model framework section, I find it hard to understand the network architecture that the authors used in this work.
I think it would be benefit to include a table detailing the network architecture.
In addition, in figure 3, I think the description of the figure could be revised to include more details such as: CNN kernel size, what the solid white arrows mean, what the dashed red arrow means, etc.
The authors do not mention the architecture of the RNN, LSTM, GRU that they used in this work. I think it would improve the clarity if they were included here.

**Reply:** Thank you for your valuable advice. We added an explanation of the arrows in Figure 2 (Figure 3). Among them, the gray filling arrow represents the TimeDistributed layer that is applied to a series of tensors in the processing of the time dimension. The black solid arrow means the multidimensional tensor into a one-dimensional vector. The dashed black arrow represents the fully connected layer in the network framework, and the dashed red arrow means the merging of multiple vectors into one vector. The figure is revised as follows:

[Figure]

**Figure 3: The model framework and network structure of GRU_CNN.**

In addition, In order to show the network framework more clearly and facilitate readers reading and understanding, we added Table 1 to list the input and output size of each layer in the network framework, including convolution kernel size, stride, and channel number.

**Table 1 Each layer architecture of the GRU_CNN**

| Layers | Kernel Size | Stride | Channel | Input Size | Output Size |
| --- | --- | --- | --- | --- | --- |
| Conv_uv | 7×7 | 2 | 8 | 21×21 | 8×8 |
| MaxPool_uv | 4×4 | 4 | 16 | 8×8 | 2×2 |
| Flatten _uv | - | - | 16 | 2×2 | 64 |
| Dense_uv_1 | - | - | - | 64 | 128 |
| Dense_uv_2 | - | - | - | 128 | 32 |
| Conv_sst | 7×7 | 2 | 1 | 21×21 | 8×8 |
| MaxPool_sst | 4×4 | 4 | 8 | 8×8 | 2×2 |
| Flatten_sst | - | - | 8 | 2×2 | 32 |
| Dense_sst_1 | - | - | - | 32 | 128 |
| Dense_sst_2 | - | - | - | 128 | 32 |
| Conv_p | 14×25 | 4 | 3 | 46×81 | 9×15 |
| MaxPool_p | 5x11 | 4 | 16 | 9×15 | 2×2 |
| Flatten _p | - | - | 16 | 2×2 | 64 |
| Dense_p_1 | - | - | - | 64 | 128 |
| Dense_p_2 | - | - | - | 128 | 32 |

| | | | | | |
|---|---|---|---|---|---|
| GRU_1 | - | - | - | 8×11 | 8×128 |
| GRU_2 | - | - | - | 8×128 | 128 |
| Dense_GRU | - | - | - | 128 | 32 |
| Concat_layer | - | - | - | - | 128 |

Lines 258-264 (Lines 364-370): "Three types of recurrent neural networks (RNN, LSTM, GRU) are used to train samples with eight timestamps and 11 features selected by the random forest method, according to their importance; the results of analyzing 49 TCs in 2019–2021 are then evaluated. We set the value of the batch size to 64 and the epoch to 100 and found that the model performed best when the number of neurons in the hidden layer is set to 128; this was determined via experiments using different numbers of neurons in the hidden layer. Early stopping is used to prevent overfitting. When the performance of the model in the validation set begins to decline, training is stopped to avoid overfitting due to continued training." introduce some detailed descriptions of the three recurrent neural network frameworks. Firstly, RNN, LSTM, and GRU are all recurrent neural networks with similar structures and the parameters of the three networks are the same. Secondly, their architectures are actually included in the overall framework, which is a part of our proposed model GRU_CNN, so it is not highlighted. In the case of only inputting trajectory features, these three networks are used to compare which result is better, and then put it into our fusion model.

In the discussion of table 3 (L384-L392), the authors claim that the influence of SST and geopotential height gradually increases at long-term forecasts. Can the authors provide more explanation of why this is the case?

**Reply:** This conclusion is based on the statistically average value in Table 4 (Table 3), which can not represent each tropical cyclone and is regarded as the results of the whole. It is shown that as the forecast time increases, the proportion of the steering flow to the predicted value gradually weakens, so the sea surface temperature and geopotential height increase accordingly. However, we have not found relevant literature to explain this phenomenon. Meteorologically, sea surface temperature will drive the TCs to the warm sea surface, so it will not affect the movement of the TCs in a short time. The geopotential height represents the weather system at high and low altitudes, and it will affect the movement of TCs for a long time. When there is a subtropical anticyclone staying in the north of the TC, it will cause the cyclone to stagnate or move slowly, which involves the analysis of weather patterns in meteorology. It is very interesting, but beyond the scope of this manuscript.

Since the authors compare the performance of GRU_CNN with other methods: FAXAI, MITAG, and IN-FA in figures 7-9, I think it would be more convincing if the authors can also provide detailed comparison between these models like in the table 3.

**Reply:** Here, FAXAI, MITAG, and IN-FA are all the TCs' names, not the model method. We selected three of the 54 typhoons in the test sets and analyzed them. Figure 5-7 (Figure 7-9) shows their actual and predicted paths. The three TCs all have the common characteristics of track turning and the other 51 TCs' forecast tracks are shown in the supplementary document.

Specific comments:

L79: missing a space between a reference and the word "applied"
**Reply:** Suggestion adopted.

Figure 4: this figure could instead show the difference between the predicted longitudes/latitudes with the observed longitudes/latitudes to improve clarity and readability.
All figures' texts and labels can be a bit bigger to improve readability.
**Reply:** The RMSE for the test sets for the three recurrent neural networks have been listed in Table 2 (Table 1). In addition, the comparison of the three recurrent neural networks is not the focus of this manuscript, and figure 4 does not show a good contrast relationship, thus it is deleted after careful consideration.
Suggestion adopted. After careful examination, we adjusted the size of the text in all figures.

L397: what are these methods: FAXAI, MITAG, IN-FA? Can you provide a short description and references for these methods?
**Reply:** This question has been answered above in the fourth General comment.

---

## Author Response (AR2)

Dear Dr. Kieu

We would like to thank you for the opportunity to revise and resubmit our manuscript titled as "Forecasting tropical cyclone tracks in the Northwest Pacific based on a deep-learning model". It is a great honor to have your recognition of this work. We found the reviewers' comments to help revise the manuscript and have carefully considered the suggestions.

We also included a response in which we addressed comments the editor made in the following (in blue after the corresponding comments). We hope that these modifications can fulfill the requirements to make the manuscript acceptable for publication. I confirm that all authors listed on the manuscript concur with the submission in its revised form. Should you have any remaining questions, I will be happy to address them.

**Editor comments:**
**Dear Dr. Wang,**

**Thank you for your revision. Based on the Reviewers' comments, I agree that your manuscript is close to final acceptance. Please consider further correcting the Figure 2 caption and legend for clarity as suggested by Reviewer 2. We look forward to receiving your final revision and thank you again for submitting your work to GMD.**

**With Regards,**
**Chanh Kieu, topical editor.**
Reply: We improved the figure presentation for clarity and ensured that the caption in the figure is consistent with the text. The specific additions and modifications are as follows:
(1)    "Timedistributed" in the figure was changed to "TimeDistributed".
(2)    The legend of "Dense" was modified to an arrow filled with slashes, and Lines 243-244 in the manuscript were modified accordingly.
(3)    To better help describe the processes and understand the structure of the model, we added a 2D tensor input, a 3D tensor input, and a 1D vector in the legend, accordingly, some adjustments were made in the manuscript.
       Lines 237-238: "that is, the area of the light gray shaded region in Fig. 2 represents 3D tensor input layers of the CNN model " was added.
       Lines 246-247: "The area of the dark gray shaded region Fig. 2 is the two-dimensional trajectory data of the TCs (2D tensor input layer) " was added.
       Line 242: "(1D vector)" was added.

Thank you again for your consideration of the revised manuscript.

Sincerely,

Liang Wang with a grateful heart
Address: Institute of Atmospheric Physics, Chinese Academy of Science
Beijing, China

Email: wangjing202@mails.ucas.ac.cn

March 22, 2023